# Transcriptomic Signatures of the Foetal Liver and Late Prenatal Development in Vitrified Rabbit Embryos

**DOI:** 10.3390/vetsci11080347

**Published:** 2024-08-01

**Authors:** José Salvador Vicente, Jesús Valdés-Hernández, Francisco Marco-Jiménez

**Affiliations:** Instituto de Ciencia y Tecnología Animal, Universitat Politècnica de València, 46022 Valencia, Spain; jvicent@dca.upv.es (J.S.V.); jvalher1@upvnet.upv.es (J.V.-H.)

**Keywords:** assisted reproduction technologies, vitrification, transcriptome, liver, foetus, ultrasonography, rabbit

## Abstract

**Simple Summary:**

Assisted reproduction technologies (ARTs) are usually safe; however, recent evidence suggests we need to look at potential risks in adulthood for better safety. ART techniques, like embryo vitrification, differ from natural conditions, which can potentially impact foetal development and life after birth. This study examined whether hepatic changes previously described after birth are already present in foetal livers at the end of gestation. We performed a comparison of phenotype and hepatic genome-wide mRNA expression via RNA sequencing between fresh and vitrified transferred rabbit embryos. As a result, we found phenotypic differences at 24 days of gestation, with vitrified embryos having lower foetal and liver weights and shorter body lengths. Moreover, offspring derived from vitrified embryos tended to be heavier, indicating a growth spurt in the last week of gestation. Additionally, only a total of 12 differentially expressed genes (DEGs) were detected among foetus groups, some of which are known for their role in lipid metabolism and the stress and immune response. Therefore, our results suggest that vitrification and embryo transfer manipulation induce an adaptive response in embryos and foetuses, which is apparent in the hepatic tissue at the end of the gestation period.

**Abstract:**

Assisted reproduction technologies (ARTs) are generally considered safe; however, emerging evidence highlights the need to evaluate potential risks in adulthood to improve safety further. ART procedures like rederivation of embryos by vitrification differ from natural conditions, causing significant disparities between in vitro and in vivo embryos, affecting foetal physiology and postnatal life. This study aims to investigate whether hepatic transcriptome and metabolome changes observed postnatally are already present in foetal livers at the end of gestation. This study compared fresh and vitrified rabbit embryos, finding differences between foetuses obtained by the transfer of fresh and vitrified embryos at 24 days of gestation. Rederived embryos had reduced foetal and liver weights and crown-rump length. However, the offspring of vitrified embryos tended to be born with higher weight, showing compensatory growth in the final week of gestation (59.2 vs. 49.8 g). RNA-Seq analysis revealed 43 differentially expressed genes (DEGs) in the foetal liver of vitrified embryos compared to the fresh group. Notably, downregulated genes included BRAT1, CYP4A7, CYP2B4, RPL23, RPL22L1, PPILAL1, A1BG, IFGGC1, LRRC57, DIPP2, UGT2B14, IRGM1, NUTF2, MPST, and PPP1R1B, while upregulated genes included ACOT8, ERICH3, UBXN2A, METTL9, ALDH3A2, DERPC-like, NR5A2-like, AP-1, COG8, INHBE, and PLA2G4C. Overall, a functional annotation of these DEGs indicated an involvement in lipid metabolism and the stress and inflammatory process or immune response. Thus, our results suggest that vitrification and embryo transfer manipulation induce an adaptive response that can be observed in the liver during the last week of gestation.

## 1. Introduction

While assisted reproduction technologies (ARTs) are generally regarded as safe, emerging evidence underscores the importance of evaluating potential risks in adulthood to further enhance safety [1,2,3,4]. ART procedures, such as in vitro culture and cryopreservation, deviate from in vivo conditions, leading to notable disparities between embryos developed in vitro and those in vivo [5,6]. In vitro production technologies involve the manipulation of oocytes and/or embryos during a critical period of mammalian development characterized by significant epigenetic changes [7,8,9,10,11]. Depending on their nature, these manipulations can impact epigenetic reprogramming, subsequently influencing individual outcomes [7,9,11,12].

There is mounting evidence suggesting that cellular manipulations during standard ART procedures have substantial implications at the gene expression and metabolomic levels in embryos [10,13,14]. During early embryonic development, the maternal environment supports optimal embryogenesis by providing stage-specific nutrients, growth factors, hormones, antioxidants, and other regulatory molecules necessary for the embryo’s evolving developmental and metabolic requirements [15,16,17]. Increasing evidence indicates that early exposures to stressful situations that disrupt these meticulously orchestrated milestones may influence developmental trajectories in the short and long term [13,14]. Under suboptimal conditions, the embryo’s developmental plasticity may compensate for environmental challenges. García-Domínguez et al. [18] reported a significant reduction in certain metabolites involved in metabolic pathways, such as the Krebs cycle, amino acids, unsaturated fatty acids, and arachidonic acid metabolism.

Cryopreservation, a technique widely used in both livestock breeding and human medicine, involves substantial cellular manipulation. This process includes in vitro handling, exposure to cryoprotectants, refrigeration, freezing prior to immersion in nitrogen, and warming and washing before transfer. These processes impose harsh conditions on embryos, influencing gene expression and subsequently affecting the proteome, metabolome, and phenotype of adult animals [14]. Studies in rabbits have linked cryopreservation to short- and long-term phenotypic changes, including variations in growth, body weight, and liver function [18,19]. Interestingly, by the late stages of gestation, no differences were observed by late gestation in foetal and placental weights, despite the detection of transcriptomic and proteomic changes in the placenta [20,21]. Moreover, a higher incidence of epigenetic disorders has been demonstrated in ART-conceived rabbits and subsequent generations [12]. This highlights the potential long-term implications of ART techniques.

The liver, a vital organ in metabolism and hormonal regulation, exhibits changes in ART-conceived animals [12,18,22]. These animals often present slightly lower serum insulin-like growth factor-I (IGF-I) and cholesterol levels, possibly due to accelerated growth compared to that of naturally conceived counterparts [22,23,24]. Additionally, low cholesterol levels in ART animals may be associated with the downregulation of the APOA4 gene, a key molecule involved in cholesterol efflux [22,25,26]. Meanwhile, metabolomic and transcriptomic analysis of prepubertal liver suggests weakened zinc and lipid metabolism (fatty acids and steroids) in vitrified-conceived rabbits and across subsequent generations, which could be associated with phenotype changes in the liver, heart, and body weight, albeit without effects on reproductive performance and health status [12,18].

Thus, there is concern not only about whether the cryopreservation process could induce changes in preimplantation embryos but also whether it could alter foetal physiology and, consequently, postnatal life. In this regard, the present study aimed to investigate whether the changes observed in the hepatic transcriptome and metabolome profiles during the postnatal stage are already apparent in the livers of foetuses at the end of gestation.

## 2. Materials and Methods

All chemicals in this study were purchased from Sigma–Aldrich Química S.A (Madrid, Spain), unless stated otherwise.

### 2.1. Animals

Rabbit does belonging to the New Zealand White line from the ICTA (Institute of Animal Science and Technology) at the Universitat Politècnica de València (UPV Valencia, Spain) were used as donors and recipients. All the experimental procedures used in this study were performed in accordance with Directive 2010/63/EU EEC for animal experiments and reviewed and approved by the Ethical Committee for Experimentation with Animals of the Universitat Politècnica de València, Spain (research code: 2018/VSC/PEA/0116).

### 2.2. Embryo Recovery

Forty donor does received an intramuscular injection of 12.5 IU of eCG (0.5 mL Cuniser500, Hipra, Girona, Spain) 48 h before artificially insemination (AI) with pooled sperm from fertile males at a rate of 20 × 106 spermatozoa. Only ejaculates exhibiting a white colour and possessing more than 70% of motility rate and less than 20% of abnormal sperm were pooled and used to inseminate. Motility was examined using a computer-assisted sperm analysis system (ISAS, Proiser SL, Paterna, Spain) from a pool sample diluted 1:20 in an AI extender and placed into a Makler counting chamber (Sefi Medical Instruments, Haifa, Israel) at 37 °C. Concentration and morphology evaluations were carried out with a 20 µL aliquot diluted 1:20 with 0.25% glutaraldehyde solution in a Thoma chamber and via a phase contrast at a magnification of 400×. Immediately after insemination, ovulation was induced by intramuscular injection of 1 μg buserelin acetate (Receptal, MSD Animal Health, Salamanca, Spain). Donors were euthanized at 72 h post-insemination, and embryos were collected at room temperature by flushing the oviducts and the first one-third of the uterine horns with 5 mL of embryo recovery media consisting of Dulbecco’s phosphate-buffered saline (DPBS) supplemented with CaCl2 (0.132 g/L), 0.2% (*w*/*v*) bovine serum albumin (BSA), and antibiotics (HyClone Penicillin-Streptomycin Solution. Penicillin 10,000 IU/mL and streptomycin 10,000 μg/mL, Fischer Scientific, Madrid, Spain). After recovery, morphologically normal embryos (morulae) with a homogeneous cellular mass, mucin coat, and intact zona pellucida (according to International Embryo Transfer Society classification) were distributed in pools of about 15 embryos for a vitrification procedure or fresh transfer.

### 2.3. Vitrification and Warming Procedure

Vitrification procedure was carried out in two steps at 20 °C. In the first step, embryos were placed for 2 min in a vitrification solution consisting of 12.5% dimethyl sulphoxide (DMSO) and 12.5% ethylene glycol (EG) in DPBS supplemented with 0.2% of BSA. In the second step, embryos were suspended for 30 s in a solution of 20% DMSO and 20% EG in DPBS supplemented with 0.2% of BSA. Then, embryos suspended in vitrification medium were loaded into 0.25 mL plastic straws, and two sections of DPBS were added at either end of each straw, separated by air bubbles. Finally, straws were sealed and plunged directly into liquid nitrogen. Warming was performed by horizontally placing the straw 10 cm from liquid nitrogen for 20–30 s; when the crystallization process began, the straws were immersed in a water bath at 20 °C for 10–15 s. The vitrification medium was removed while loading the embryos into a solution containing DPBS and 0.33 M sucrose for 5 min, followed by one bath in a solution of DPBS for another 5 min.

### 2.4. Embryo Transfer by Laparoscopy

A total of 281 vitrified and 198 fresh morphologically normal embryos were transferred into oviducts by laparoscopy to 36 recipient does (15 to 16 to vitrified embryos and 10 to 12 fresh embryos by recipient doe) following the procedure described by Besenfelder and Brem [27] and García-Domínguez [28]. Ovulation was induced in recipient does with an intramuscular dose of 1 µg buserelin acetate (Receptal, MSD Animal Health, Salamanca, Spain) 68–72 h before the transfer. The equipment used was a Hopkins^®^ Laparoscope, which is a 0° mm straight-viewing laparoscope, 30 cm in length, with a 5 mm working channel (Karl Storz Endoscopia Ibérica S.A., Madrid, Spain). To sedate the does during laparoscopy, anaesthesia was administered by an intramuscular injection of 5 mg/kg of xylazine (Bayer AG, Leverkusen, Germany), followed 5–10 min later by an intravenous injection into the marginal ear vein of 6 mg/kg of ketamine hydrochloride (Imalgène, Merial SA, Lyon, France). During laparoscopy, 3 mg/kg of morphine hydrochloride (Morfina^®^, B. Braun, Barcelona, Spain) was administered intramuscularly. After transfer, does were treated with antibiotics (4 mg/kg of gentamicin every 24 h for 3 days, 10% Ganadexil, Invesa, Barcelona, Spain) and analgesics (buprenorphine hydrochloride: 0.03 mg/kg every 12 h for 3 days, Buprex^®^, Esteve, Barcelona, Spain; meloxicam: 0.2 mg/kg every 24 h for 3 days, Metacam^®^ 5 mg/mL, Norvet, Barcelona, Spain).

### 2.5. Foetal Growth by Ultrasound Examination

Six recipient does from each experimental group were examined at days 24, 26, and 28 post-ovulation induction by a portable colour Doppler ultrasound device (Esaote, Barcelona, Spain) with a 7.5 MHz linear probe (4–12 MHz range). Prior to examination, does were sedated by intramuscular injection of 35 mg/kg of ketamine and 5 mg/kg of xylazine, and the abdomen of the doe was clipped. Does were placed in dorsal recumbency in a polystyrene cage where they were prevented from moving and where the ultrasound scanning gel was applied. The ultrasound examination was performed transcutaneously from right to left with the probe in a sagittal orientation and, after localization of different foetal sacks, the identifiable structures (foetus and foetal and maternal placenta) were measured from frozen frame pictures on the monitor, using the Esaote 16 ultrasound software. For the foetal sack surface (FS), the measurement was taken when the largest surface area appeared on the screen. For whole foetus measurements, crown-rump length (CRL) was determined as the maximum distance from crown to tail basis with the foetus, and cross-section of the foetus was performed in the upper part of the abdomen when the heart was observed. Placental measurements were determined when the maximal placental surface with the two-lobed foetuses being visible on the screen [21,29,30]. No more than 6 foetuses per recipient doe and from 22 to 32 foetuses by embryo group type were measured at each day of gestation.

### 2.6. Foetal and Offspring Phenotypic Determinations

Ten recipients were euthanized at 24 days of gestation to obtain samples for liver transcriptome analysis and to determine the foetal body, liver, placenta, and foetal weights. Additionally, the hepatosomatic index, defined as the ratio of liver weight to total body weight, was calculated for each foetus. For the remaining recipient does, birth weights were recorded.

### 2.7. RNA Isolation, Sequencing, and Analysis of Foetal Liver

Total RNA was extracted from 10 foetal livers at day 24 of gestation. Immediately, samples were washed with phosphate-buffered saline (PBS) to remove blood remnants and stored in RNA later (Ambion Inc., Huntingdon, UK) at −20 °C until analysis. Total RNA was isolated from liver samples using a combination of mirVana^TM^ kit (Ambion, Austin, TX, USA) and AllPrep kit (Quiagen, Hilden, Germany) following the manufacturer’s recommendations. The resulting RNA was tested for quality and quantity using the Agilent 2100 Bioanalyzer System (Agilent Technologies, Santa Clara, CA, USA), and samples with an RNA integrity number (RIN) greater than 8 and greater than 3 µg of total RNA were selected for the sequencing experiment. Subsequently, total RNA samples were submitted to the company Macrogen (Seoul, Republic of Korea) for library preparation and sequencing. Libraries from each of the 10 samples were generated, and 5 individual foetal liver transcriptomes of each experimental group (vitrified: V and fresh: F) were sequenced using an Illumina Hiseq-2000 sequencer (Illumina, San Diego, CA, USA). Raw sequences files are available at the NCBI Sequence Read Archive (BioProject ID: PRJNA1115099, Appendix A).

Raw FASTQ files were submitted for processing and analysis according to the NASA GeneLab RNA sequencing (RNA-Seq) consensus pipeline [31]. Briefly, this standardized pipeline consists of several steps: (1) quality control using FastQC v0.12.1 software [32]; (2) read trimming using Trim Galore! v0.6.10 [33]; (3) alignment against the reference genome “UM_NZW_1.0 (RefSeq GCF_009806435.1)” (*Oryctolagus cuniculus*) using STAR v2.7.11a software [34]; and (4) gene expression quantification and mapping to transcriptome sequences using RSEM v1.3.3 [35]. In brief, this study produced 2 × 100 bp reads, an average of 37.52 million paired-reads per sample, and 82.41% (range from 70.80 to 85.40%) of uniquely mapped reads.

### 2.8. Statistical Analysis

#### 2.8.1. Phenotypical Parameter Analysis

Analyses of phenotypic parameters were performed with SPSS statistical software package (v23.0, SPSS Inc., Chicago, IL, USA, 2002). Phenotypic parameters such as foetal body, liver, and foetal and maternal placenta weights, crown-rump length, cross-section of foetus, and maternal and foetal placenta surfaces were analysed by a general linear model (GLM). The model included the fixed factors of embryo type (fresh or vitrified) days of gestation (24, 26 and 28th day) for ultrasound parameters, with recipient does as a random effect and the litter size at birth as a covariate.

Additionally, the effects of vitrification on survival rates (foetal at day 24 and offspring at partum) were analysed using a GLM with the embryo type (fresh or vitrified) as a fixed factor. The error was modelled with a binomial distribution using the probit link function. Binomial data were assigned a value of one if positive development was achieved or a zero if it was not. *p*-value ≤ 0.05 was considered significant.

#### 2.8.2. Differential Gene Expression Analysis in Foetal Liver

Identification of DEGs was conducted using DESeq2 v1.42.0 R package [36]. The read count matrix (31,346 genes in total) was filtered, excluding all genes with < 10 reads via rowSums function, and 18,104 genes were finally retained. The median of ratios method implemented in the DESeq2 package was used to conduct gene count normalization. The differences between groups in gene expression were determined by applying a negative binomial model, including the group (2 levels, F and V) as a fixed effect. The Wald test was used to identify DEGs between groups, and the raw *p*-values were corrected for multiple testing using the Benjamini–Hochberg (BH) method. Then, DEGs with an BH adjusted *p*-value < 0.05 (Padj) and a log_2_ fold change (FC) threshold of at least 0.58 (|absolute FC of 1.5|) were considered for further DEG data exploration and functional analysis.

ClustVis software (v2.0, https://biit.cs.ut.ee/clustvis/ (accessed on 25 July 2024)) was used to perform the principal component analysis (PCA) of expression data and heat map clustering [37]. In addition, we used a complementary heatmap via the ComplexHeatmap v2.14.0 package [38] to illustrate the different clusters of variables. Functional annotation of DEGs, enrichment analysis of their associated gene ontology (GO) terms (molecular function, biological processes and cellular components), and KEGG pathways analysis were conducted using the Bioinformatic software DAVID tool (Database for Annotation, Visualization and Integrated Discovery (version 6.8; http://david.abcc.ncifcrf.gov accessed on 25 July 2024)). Significant GO terms and pathways were considered at a *p*-value of <0.05.

The Pearson correlation between the normalized gene expression levels (i.e., with normalized function = TRUE of DESeq2 package) of DEGs and the phenotypic parameters, including foetus and placenta weights, as well as ultrasound measurements such as foetus length, cranial dimensions, and maternal and foetal placenta thickness, were analysed.

## 3. Result

### 3.1. Prenatal Survival and Development of Fresh and Vitrified Embryos

Of a total of 49 fresh and 65 vitrified transferred embryos, 41 and 43 (0.84 ± 0.053 vs. 0.66 ± 0.067, *p*-value < 0.05) developed at 24 days of gestation, respectively. Regarding phenotypic parameters, both foetal and liver weights were affected in foetuses derived from the vitrification process. However, the hepatosomatic index and placental weights (foetal and maternal) showed no significant differences between groups (Table 1).

The crown-rump length of foetuses was the only parameter showing significant differences on the 24th day of gestation (65.4 ± 1.65 vs. 58.4 ± 1.52 mm, for fresh vs. vitrified embryos, respectively, Figure 1). The rest of the foetal parameters measured did not show differences regarding gestation between the embryo group (Figure 1).

### 3.2. Postnatal Survival Rate and Body Weight at Birth

Postnatal survival rate and pup weight at birth showed significant differences between embryo groups. Survival rate was 0.71 ± 0.058 and 0.47 ± 0.055 (*p*-value < 0.05) for fresh and vitrified transferred embryos, respectively, and pups derived from vitrified transferred embryos were heavier than fresh ones (59.2 ± 2.55 and 49.8 ± 2.55, respectively, Table 2).

### 3.3. Differential Gene Expression and Functional Analysis in Foetal Liver

Results from RNA-Seq analysis revealed that of the 18.104 gene expressions identified, 43 were DEGs (FC > 1.5 and Padj < 0.05) in the livers of foetuses derived from fresh embryo (F group) compared to vitrified embryos (V group). The volcano plot shows the statistical significance of the 43 DEGs, including 30 downregulated (red points) and 13 upregulated (green points) genes (Figure 2a). Consistent with the results, downregulated genes such as LOC100338464 (RPL22L1), LOC127482602, LOC100357982, LOC127486131 (DIPP2), LOC127489097 (IFGGC1), LOC127489095 (IRGM1), UGT2B14, LOC100352702, LOC127491078 (MPST), and LOC127489092, and upregulated genes such as LOC127486773 (ALDH3A2), PLA2G4C, ERICH3, LOC103348263 (DERPC-like), LOC100338442 (EEF1AKMT1), LOC127489132 (COG8), LOC103347040, LOC100353617 (NR5A2-like), INHBE, and LOC100349068 (AP-1) were the top 10 exhibiting the greatest significant difference in expression (i.e., absolute FC ≥ 3.50 and Padj < 0.05) between the V and F groups (Figure 2a and Appendix A).

The heatmap classified the 10 samples into two different main clusters corresponding to both groups analysed (fresh and vitrified) and divided the 43 DEGs into three main clusters, two of them to downregulated expression genes and the rest to upregulated expression genes in liver foetuses derived from vitrified embryos (Figure 2b). Of these, two clusters grouped the downregulated gene expression (n = 8 and 22 genes with red colour scale of FC, respectively), while the third cluster grouped the upregulated expression gene expression (n = 13 genes with green colour scale of FC). For instance, three of the aforementioned 10 downregulated genes were identified in cluster 1, but other genes with similar expression profiles were also included (e.g., PPP1R1B, LOC100357720, UGT2B4, KEF51_p09, or ATP8). However, cluster 2 was composed of seven members of the top 10 for downregulated genes, as well as several genes with significant transcriptomic changes (e.g., LRRC57, BRAT1, CYP4A7, CYP2B4, and DCPS). Likewise, cluster 3 was composed of top 10 upregulated genes, including three other genes, i.e., LOC127488823, ACOT8, and UBXN2A.

In addition, a functional analysis of DEGs using the DAVID cluster approach revealed that the main functional terms affected (FDR < 0.05) corresponded to “GO:0035458~cellular response to interferon-beta”, “IPR007743:Immunity-related_GTPase-like”, “DOMAIN:IRG-type G”, “IPR030385:G_IRG_dom”, “GO:0098542~defence response to other organism”, “GO:0006952~defence response”, and “GO:0005789~endoplasmic reticulum membrane”. However, a total of 21 terms were not clustered (Appendix A).

### 3.4. Relationships of the Gene Expression with Phenotypical Traits, Including Foetal-Placental Measurements

Pearson’s correlations between gene expression levels of DEGs and phenotypic traits are reported in Figure 3. Remarkably, strong significant correlations (|absolute r of 0.7|) were found between DEGs and liver weight.

Most of the DEGs that were positively correlated with liver weight were downregulated in foetuses derived from vitrified embryos, such as LOC100356896 (or RPL22L1), LOC100339459 (or PPILAL1), LRRC57, LOC100338464, LOC100354232 (or A1BG), BRAT1, CYP4A7, LOC127489097 (or IFGGC1), and LOC127486131 (or DIPP2). Additionally, a large number of DEGs that correlated with liver weight were upregulated in foetuses derived from vitrified embryos, such as LOC127489132 (or COG8), UBXN2A, LOC100353617 (or NR5A2-like, LOC100338442 (METTL9 or EEF1AKMT1), LOC1127486773 (or ALDH3A2), LOC103348263 (or DERPC-like), and LOC100349068. Some of these genes also showed moderate to strong significant correlations with foetal body weight, hepatosomatic index, and crown-rump length at 24 days of gestation (Figure 3).

## 4. Discussion

We found evidence of specific variations in 24-day-old foetuses (late gestation in a rabbit model) derived from vitrified embryos in a stimulated ART cycle, including differences in survival, body weight, liver weight, and foetal length. However, body weight did not differ at birth. Additionally, we described gene expression changes in the liver of these 24-day-old foetuses in response to embryo vitrification manipulation. We selected the liver for gene expression studies because it exemplifies an organ with diverse roles in mammalian function throughout different life stages. The liver is a highly metabolic organ performing numerous essential functions, such as defending against xenobiotics, aiding in digestion and in the metabolism of absorbed nutrients, regulating blood lipids, synthesizing a wide array of secreted proteins, and maintaining a hormonal balance [39].

In recent years, the rabbit has been widely used as a model to demonstrate the applicability of embryo cryopreservation for the re-establishment of populations [12,18,19,29,30,40]. While the efficacy of embryo cryopreservation has been demonstrated, several studies have directly tested that embryos exposed to these procedures induce molecular, metabolic, and ultra-structural disruptions that affect early foetal and placental development [20,21,22,41]. Using transcriptome and proteome analyses of the foetal placenta at 14 days of gestation (early implantation), Saenz-de-Juano et al. [20] found profound differences between vitrified and fresh transferred conception foetuses that could be implicated in gestational losses, showing biomarkers similar to those observed in pre-eclampsia placentas and cases of intrauterine growth restriction [42,43]. Nevertheless, early embryos display a remarkable homeostatic flexibility, allowing them to regulate metabolism optimally in response to stress [44]. Therefore, a significant proportion of vitrified embryos can overcome these alterations and sustain pregnancy. We observed that the survival rate for vitrified embryos was 57% at 24 days of gestation and 47% at birth, which are significantly lower than the rates achieved with fresh embryos. These observations are consistent with several studies in this species using late morula–early blastocyst stages [18,22,40,45] and highlight that a satisfactory proportion of vitrified embryos possess sufficient plasticity to survive this stressful ART technique. However, the 24-day gestation foetuses derived from vitrified embryos exhibited less development in terms of foetal length and body weight (including liver weight). This reduced development appears to be compensated in the final week, with vitrified and fresh embryos reaching similar body weights at birth, as indicated by ultrasonography measurements taken on the 26th and 28th days of gestation [22]. Focusing on mRNA expression, we observed several genes belonging to the complement and coagulation KEGG pathway (FGG, F10, C9, FGB, SERPINF2, F2, F9, CFI, and PROC).

These results are consistent with those of previous studies in this species and at this embryonic stage, highlighting that a satisfactory proportion of vitrified embryos have sufficient plasticity to respond viably to the ART technique. Nevertheless, as measured by ultrasonography, both the length and weight of the foetus and liver were lower in those derived from the 24-day foetuses. These foetuses experienced compensatory growth in the final week, reaching a similar size to fresh foetuses, as indicated by the ultrasonography performed on the 26th and 28th days of gestation. This accelerated growth seems to continue in the first weeks of life as offspring derived from vitrified embryos weighed significantly more than those born without assisted reproduction techniques from birth and during the first weeks of age [19,22]. Nevertheless, adult body weight was lower than that achieved through natural mating [18]. Therefore, the foetal growth parameters at the end of gestation do not allow for anticipating changes in the postnatal growth pattern. Rabbits derived from vitrified embryos reach lower adult weights and could exhibit differences in the weight of significant organs such as the liver, kidney, or adrenal glands in young rabbits [19] or in the liver and heart in adult rabbits [18].

This study of the liver transcriptome of these 24-day foetus samples revealed that a total of 43 genes were differentially expressed between the vitrified (V) and fresh (F) groups. Likewise, functional analysis with DEGs mainly suggested the over-representation of pathways associated with the immune response, stress response, and steroid biosynthesis. Furthermore, when the correlation of DEGs with the phenotypic parameters was analysed, it was observed that 19 of them had a significant and high correlation (|absolute r of 0.7|) with liver and foetus weight, as well as with the hepatosomatic index. Among these genes, the following ones stand out, the downregulation of cytochrome P450 4A7 (CYP4A7) and the upregulation of nuclear receptor subfamily 5 group A member 2-like (NR5A2-like) and aldehyde dehydrogenase family 3 member A2 (ALDH3A2), with all of them being involved in lipid metabolism, steatosis, and in enhancing insulin sensitivity metabolism [46,47,48,49]. Moreover, NR5A2-like and peptidyl-prolyl cis-trans isomerase A-like (PPILAL1) have been related with inflammatory and anti-apoptotic process. This finding aligns with previous studies, which reported disturbances in the biosynthesis and metabolism of sterols, fatty acids, and lipids in the placenta, foetal liver, and adult serum and tissues after ART [4,12,20,22,24,50,51]. Specifically, variations in linoleic acid metabolism, arachidonic acid metabolism, cholesterol metabolism, steroid hormone biosynthesis, and retinol metabolism have been reported in prepuberal and adults rabbit derived from vitrified embryos. Such alterations could be associated with the modifications observed in organ growth, such as liver and body growth patterns in postnatal life [18,19,22,30]. Nevertheless, it is also assumed that vitrification procedures involve a decrease in lipid content (including unsaturated fatty acids such as arachidonic, linoleic, linolenic, oleic, and palmitoleic acids and saturated fatty acids like myristic, palmitic, and stearic acids) [52]. Indeed, it is essential to devise strategies to make up for the deficiency of such specific lipids in order to minimize cell damage and maintain the cellular stability caused by this reduction.

Another group of relevant genes related with liver weight were those that were down-expressed (PPILAL1, A1BG, LRRC57; BRAT1, RPL22L1, DIPP2 and IFGGC1) and up-expressed (METTL9 or EEF1AKMT1 and COG8). In relation to genes coding for ribosomal proteins, we observed that the 60S ribosomal protein L22-like 1 (RPL22L1) was the most significant gene (FC = −108.84) when comparing the V and F groups. Most ribosomal proteins are considered essential and static components that contribute to ribosome biogenesis and protein synthesis. However, RPL22L1 or Like1 have been reported to play critical and extraribosomal roles in embryogenesis [53]. Indeed, they antagonistically control morphogenesis through a developmentally regulated localization to the nucleus, where they modulate splicing of the pre-mRNA encoding *SMAD2*, an essential transcriptional effector of Nodal/TGF-β signalling.

Meanwhile, it is important to highlight that the peptidyl-prolyl isomerase (*PPI*) family including the peptidyl-prolyl cis-trans isomerase A-like member (also known as *PPILAL1*) catalyses the cis-trans isomerization of peptide bonds’ N-terminal to proline residues in polypeptide chains, which help to manage the transcription process during development, endocrine responses, or environmental stresses. Therefore, PPI members are implicated in various cellular processes, also including proliferation, differentiation, or inflammation [54,55,56,57,58,59,60,61,62]. The alpha-1B-glycoprotein (A1BG) has been associated with the regulation of cell division in the liver and implicated in mediating the acute-phase response during inflammation [63,64]. The BRCA1-associated ATM activator 1 (BRAT) gene deficiency has been linked to increased glucose metabolism and mitochondrial malfunction [65] and, through the BRG1 gene, negatively affects liver regeneration [57]. Furthermore, the interferon-inducible GTPase 1-like (IFGGC1) deficiency can be associated not only with decreased resistance to intracellular pathogens in the liver but also with reduced lipid peroxidation [66]. Lastly, as the liver is the organ responsible for producing most glycosylated serum proteins, malfunctions in the COG complex significantly impact processes such as Golgi integrity and glycosylation, affecting the synthesis of glycoproteins and glycolipids, which might disrupt metabolism [54,67].

In summary, the expression of these genes could affect both liver function and tissue regeneration. This altered hepatic capacity might be more relevant in the postnatal stage, where individuals are exposed to greater environmental challenges, with the detoxifying stress and immune responses becoming crucial. Monitoring the liver function and health of individuals born from vitrified embryos throughout their lives is necessary to determine the extent of these modifications. Additionally, these effects should be studied with attention to the effect of gender, as Feuer et al. [24] observed a sexually dimorphic effect of reproductive technologies such as in vitro fertilization on adult mouse liver at the metabolic level.

## 5. Conclusions

These findings suggest that the physiological stress caused by ART techniques, such as embryo vitrification and transfer, induces an adaptive response in embryos and foetuses, which can be observed in the liver during the last week of gestation. This response could persist into postnatal life, potentially conditioning the developmental trajectory in adulthood.

## Figures and Tables

**Figure 1 vetsci-11-00347-f001:**
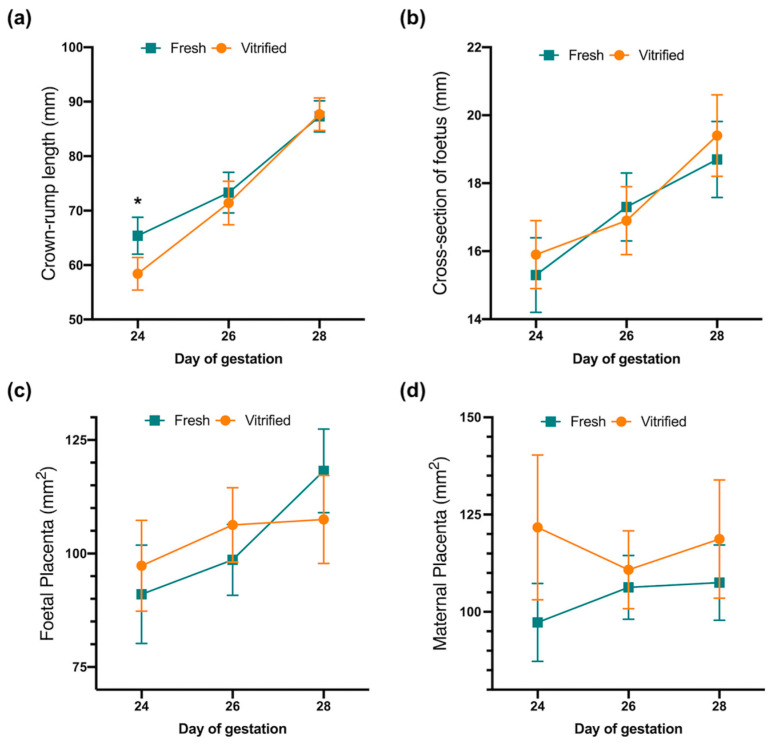
Comparison of foetal and placental growth at 24, 26, and 28 days of gestation via ultrasonography from fresh and vitrified transferred embryos. (**a**) Crown-rump length of foetus. (**b**) Cross-section of foetus. (**c**) Foetal placenta area. (**d**) Maternal placenta area. Asterisks indicate significant differences (*p*-value < 0.05).

**Figure 2 vetsci-11-00347-f002:**
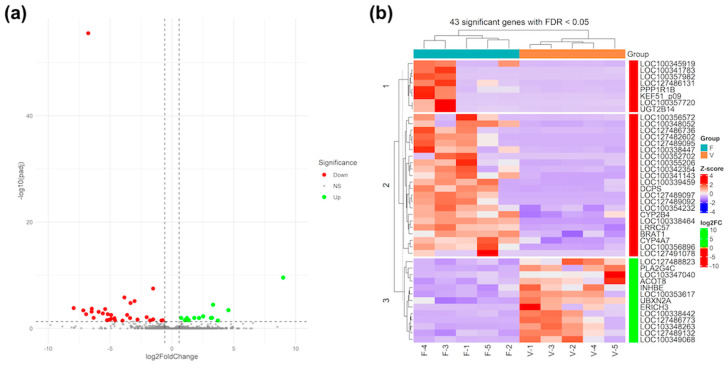
Hepatic transcriptomic analysis. (**a**) Volcano plot. The log_2_ FC indicates the ratio between gene expression level by group. (F vs. V) Each dot represents one gene. Grey dots represent non-significant DEGs between fresh group and vitrified group embryos, the green dots represent upregulated genes, and red dots represent downregulated genes. (**b**) Heat map of differentially expressed genes. Legend on the left indicate log fold change of genes. (F1–F5 = fresh embryo samples; V1–V5 = vitrified embryo samples).

**Figure 3 vetsci-11-00347-f003:**
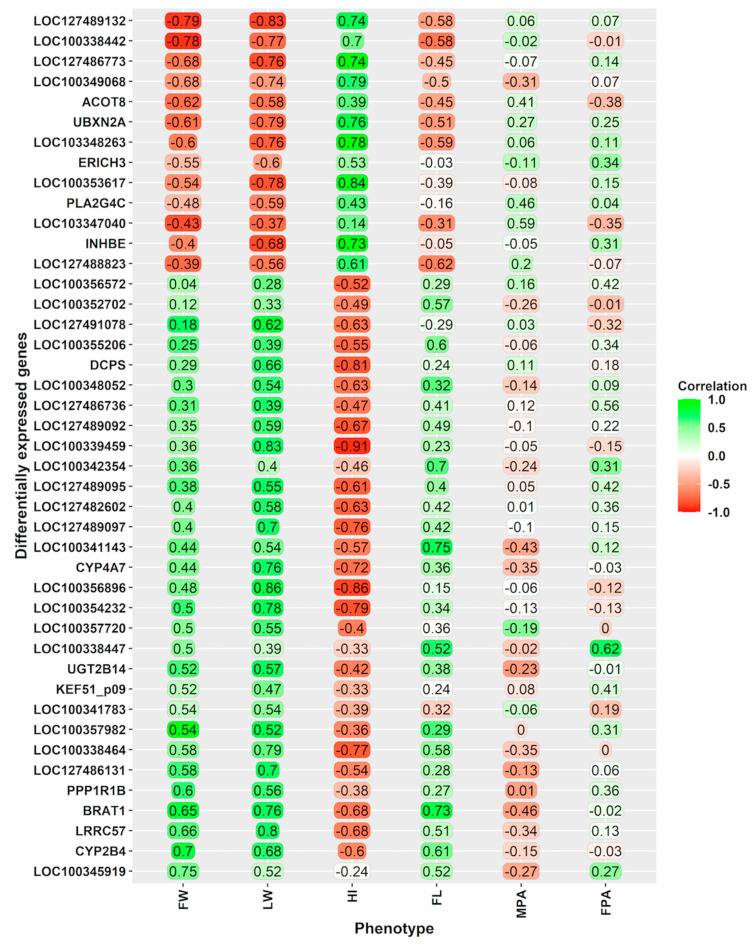
Pearson’s correlations between differentially expressed genes and foetal weight, liver weight, hepatosomatic index, and placental areas at day 24 of gestation. FW: Foetal weight. LW: Liver weight. HI: Hepatosomatic index, defined as the ratio of liver weight to total body weight. FL: Foetus length. MPA: Maternal placenta area. FPA: Foetal placenta area. The scale on the left side indicates Pearson’s correlation value (green = positively correlated, red = negatively correlated).

**Table 1 vetsci-11-00347-t001:** Prenatal survival and phenotypic parameters of fresh and vitrified transferred embryos at 24 days of gestation.

Group	n	Survival	Foetal Weight (g)			Placenta Weight (g)	
			Body	Liver	Hepatosomatic Index ^1^	Foetal	Maternal
Fresh	49	0.84 ± 0.053 ^a^	13.3 ± 0.43 ^a^	1.28 ± 0.063 ^a^	11.0 ± 0.46	2.7 ± 0.12	1.32 ± 0.103
Vitrified	65	0.66 ± 0.067 ^b^	11.4 ± 0.43 ^b^	1.06 ± 0.064 ^b^	11.8 ± 0.45	2.8 ± 0.12	1.19 ± 0.106

n: Transferred embryos. ^1^ Hepatosomatic index is defined as the ratio of liver weight to total body weight. ^a,b^ Values with different superscripts in columns differ statistically (*p* < 0.05). Data are expressed as least square mean ± standard error.

**Table 2 vetsci-11-00347-t002:** Viability and birth weight of fresh and derived transferred embryos.

Group	Embryos Transferred	Survival at Birth	Birth Weight ^1^ (g) (n)	Live Birth Weight ^1^ (g) (n)
Vitrified	146	0.47 ± 0.055 ^b^	59.2 ± 2.55 (68)	56.4 3.12 (64)
Fresh	81	0.71 ± 0.058 ^a^	49.8 ± 2.90 (56)	52.6 3.51 (51)

^1^ Value in parentheses reflects the number of data points used. Data are expressed as least square mean ± standard error. The values with different superscript letters in a column are significantly different (*p*-value < 0.05).

## Data Availability

The data presented in this study are available in the article, BioProject accession number (PRJNA1115099) and Appendix A.

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
