# Peer review of "Transcriptomic Signatures of the Foetal Liver and Late Prenatal Development in Vitrified Rabbit Embryos"

_vetsci, 2024, doi:10.3390/vetsci11080347_

Round 1

Reviewer 1 Report

Comments and Suggestions for Authors

The study aimed to evaluate the effects of embryo vitrification on foetuses at the end of gestation. Using the rabbit as an animal model, the authors conducted a comprehensive analysis of the foetal and placental growth at 24, 26, and 28 days of gestation. They then evaluated the postnatal weight, and finally, they performed a transcriptomic analysis of the liver tissue at 24 days of gestation. The findings revealed phenotypic and transcriptomic differences at 24 days of gestation that could suggest a metabolic compensatory process developed by the vitrified embryos to resolve the stress of the cryopreservation.

One of the manuscript's most significant aspects is the use of a large number of vitrified (n=281) and transferred embryos (n=198), which significantly enhances the robustness of the observations. The authors also employed ultrasound and transcriptomics to correlate phenotypical and molecular changes in the analysed foetuses. The results of this study are important for both animal breeding programs and human fertility clinics, where embryo vitrification is standard practice.

Minor comments:

Line 38-39, please review.

Line 121: I could not find equipment called the Bio-rad bioanalyzer. The nucleic acid electrophoresis from Bio-rad is called Experion™ Automated Electrophoresis System, and the Bioanalyzer is from Agilent Biosystems. Please review.

Line 193: and* greater

Figure 1a: The graph should include an asterisk or different letters on day 24 to show significant differences independently from the text.

Line 277 and line 287: It would be recommended to use the same abbreviation (Padj) throughout the manuscript. The BH-adjusted P-value can already be defined as Padj in line 232.

Figure 2b and Figure 3: I suggest adding the gene description instead of the gene symbol. That would make the figure more informative for the reader. Another possibility is to add an alternative gene symbol as the authors do in the paragraph starting in line 336. The same is true for Supplementary Table 1 on the DEGs sheet. The authors could add a column with the gene description.

In line 288, the Supplementary Table 1 is named differently, Table S1.

Line 392: Is the sentence “offspring derived to vitrified embryos” correct?

Author Response

Comment: Line 38-39, please review.

Response: We have reviewed and addressed the minor comment

Comment: electrophoresis from Bio-rad is called Experion™ Automated Electrophoresis System, and the Bioanalyzer is from Agilent Biosystems. Please review.

Response: We have reviewed and corrected the error.

Comment: Line 193: and* greater

Response: We have reviewed and corrected the mistake.

Comment: Figure 1a: The graph should include an asterisk or different letters on day 24 to show significant differences independently from the text.

Response: The asterisk has been included in the figure, with its definition added to the figure legend.

Comment: Line 277 and line 287: It would be recommended to use the same abbreviation (Padj) throughout the manuscript. The BH-adjusted P-value can already be defined as Padj in line 232.

Response: Following the suggestion, the BH-adjusted P-value has been defined as Padj.

Comment: Figure 2b and Figure 3: I suggest adding the gene description instead of the gene symbol. That would make the figure more informative for the reader. Another possibility is to add an alternative gene symbol, as the authors do in the paragraph starting in line 336. The same is true for Supplementary Table 1 on the DEGs sheet. The authors could add a column with the gene description.

Response: Thank you for the suggestion. Since the description is lengthy, it has not been included in the figures but has been provided in Supplementary Table S1 for all genes, including the DEGs.

Thank you very much for your comments. Below you will find our response to your comments and suggestions.

Comment: In line 288, Supplementary Table 1 is named differently, Table S1.

Response: It has been corrected

Comment: Line 392: Is the sentence “offspring derived to vitrified embryos” correct?

Response: It has been corrected. We change “to” by “from”.

Reviewer 2 Report

Comments and Suggestions for Authors

In this article, the authors show that embryo vitrification affects gene expression in the liver of rabbit fetuses and induces an adaptive response during the last week of gestation. In assisted reproductive technologies, operations such as cryopreservation and embryo transfer may affect the fetus and individuals born from transferred embryos, and this is a very important issue. They have found differences in the expression of genes that play a role in lipid metabolism, stress and the immune response in fetuses developed from vitrified and fresh embryos. Although further research is required to determine whether these differences affect neonatal and subsequent health, the results of this study provide useful information on the effects of embryo vitrification.

The topic addressed is meaningful, but I think there are some revisions that should be made before publication.

1)     Line 223. Please correct the chapter number “2.8.1” into “2.8.2”.

2)     Figure 1 and Table 2. Please put marks to show the significant differences.

3)     Results and Discussion. In embryo transfer experiments, the number of embryos transferred differed between groups of vitrified and fresh embryos. Is it possible that the number of implants or fetuses per recipient doe influenced the results?

Author Response

Thank you very much for your comments. Below you will find our response to your comments and suggestions.

Comment: Line 223. Please correct the chapter number “2.8.1” into “2.8.2”.

Response: It has been corrected

Comment: Figure 1 and Table 2. Please put marks to show the significant differences.

Response: It has been included

Comment: Results and Discussion. In embryo transfer experiments, the number of embryos transferred differed between groups of vitrified and fresh embryos. Is it possible that the number of implants or fetuses per recipient doe influenced the results?

Response: Based on our previous experience, we decided to use a different number of embryos transferred so that the mean number of implanted embryos in both groups would be similar and to reduce the impact of this variable. In fact, the covariate number of implanted embryos was not included in the analyses because it was not significant. However, we did include, due to their possible influence on the parameters measured on the foetuses at the end of gestation, the random effect of the recipient doe and litter size.